# What determines client satisfaction on labor and delivery service in Ethiopia? systematic review and meta-analysis

**Rekiku Fikre**[1]*, **Kidist Eshetu**[2], **Muntasha Berhanu**[3], **Akalewold Alemayehu**[4]

**1** Department of Midwifery, College of Medicine and Health Science, Hawassa University, Hawassa, Ethiopia, **2** Department of HIT, Hawassa Health Science College, Hawassa, Ethiopia, **3** Hawassa Adare General Hospital, Hawassa, Ethiopia, **4** School of Public Health, College of Medicine and Health Science, Hawassa University, Hawassa, Ethiopia

* frekiku@yahoo.com

## Abstract

### Introduction

The uptake of Health services, maternal and newborn health care outcomes are dictated by the satisfaction of clients on the service provided. Client satisfaction is one of the vital indicators to measure the quality of service. However, it is not well addressed and little evidence is existed in Ethiopia. Therefore, the purpose of this systematic review aimed to assess the prevalence and determinant of client satisfaction on labor and delivery service in Ethiopia.

### Methods

This study has included published and unpublished articles. The main databases PubMed, Embase, EBSCO, Medline, CINHAL, Poplin, and the search engine like Google and Google scholar were used from June1-30/2020. Studies with observational study design which are conducted in English language and met the eligibility criteria were included in the review. Meta-analyses with random effects were performed. Data synthesis and statistical analysis were conducted using OpenMeta and CMA version 2 software.

### Results

The pooled prevalence of client satisfaction on labor and delivery service in Ethiopia was 73.5% [95% CI [64.9%, 82.1%]. The pooled odds ratio showed a negative association between client satisfaction on labour and delivery service with Promptness of care [OR = 0.25; 95% CI: (0.18, 0.34), P = 0.0001], Free service charge [OR = 0.70; 95% CI: (0.57, 0.86), P < 0.0007], Privacy during examination [OR = 0.25; 95% CI: (0.10, 0.64), P < 0.004], Respectful maternal care [OR = 0.40;95% CI: (0.19, 0.83), P = 0.01], Plan to delivered at health facility [OR = 0.49; 95% CI: (0.37, 0.66), P < 0.00001] and ANC follow-up [OR = 0.39; 95% CI: 0.24, 0.63, P < 0.0001].

**Data Availability Statement:** All relevant data are within the paper and its Supporting Information files.

**Funding:** The authors received no specific funding for this work.

**Competing interests:** We declared that no competing interests exist.

**Abbreviations:** CMA, Comprehensive meta analysis; PRISMA, Preferred Reporting Items for Systematic Reviews and Meta-Analyses; JBI, Joanna Briggs Institute.

## Conclusions

This review revealed that client satisfaction on labor and delivery service in Ethiopia was 73.5%. Besides poor care of providers on the antepartum, intrapartum and lack compassionate and respectful care affects client satisfaction on labor and delivery service in Ethiopia.

## Introduction

Client satisfaction is defined as individuals/family perception of the care they received in line with positive outcome of the service [1]. The motives of measuring patient satisfactions consist of not only designates of the health care service from the side of patient's perspectives, but it also measures the process of the service utilization and evaluation of the care towards patient satisfaction. The concepts of quality of care is operationalized in to different dimension based on the outcome of interest. Recently the focus of the health care system is shifted to the pillar of quality of the service based on the key performance indicators giving much emphasis on client satisfaction. Client satisfaction is one of the vital indicators to measure the quality of service they received. However, it is measured based on the value and feedback of the client and family members based on the perceived care they are provided [2–4]. Maternal and neonatal mortality is still high in developing countries despite the advancement of technology, the expansion of health care facilities, and policy improvement to alleviate preventable causes of maternal and neonatal mortality [5].

The lives of the mothers and newborns were determined by lack competent providers as well as lack of compassionate and respectful care. Maternal and child health service utilization has been affected by satisfaction of the mothers during labor and delivery service [6, 7].

Globally widespread attention is given on raising the coverage of life-saving intervention for mothers and babies to meet Sustainable Development Goals (SDGs) by 2030 and the Ethiopian government has made several efforts over the past decades to decrease maternal and neonatal mortality; though the efforts were not attained the expected result as evidenced in the Millennium Development Goal (MDG) target of Maternal Mortality rate (MMR) of 267 per 100,000 births vs 420 per 100,000 births [8–10].

In developing countries, including Ethiopia several strategies were exercised. Among these, increasing the coverage of skilled birth attendant and the expansion facility-based maternity care as a means of reduction of maternal mortality can be mentioned. Yet, the quality of care is still poor due to various reasons like lack of equipment, shortage of manpower and gap of providers treating clients with companionate and respectful care [11].

There is substantial improvement in access to essential health services, nevertheless, emotional aspect of clients is ignored and leads them to poor satisfaction and low facility utilization attributed to high levels of maternal and child mortality [12, 13].

Several studies show that most women had faced so many problems during labor and delivery including insulting and lack of compassionate and respectful care by care providers. Similarly, the rights of the women were diminished and the practice of the provider become unethical [14–19].

Even though perceived satisfaction of clients depends on the outcome of the mother and neonate, this alone did not determine the quality of service [20, 21]. The perceived satisfaction of clients has a positive impact to increase service uptake, in parallel to these dissatisfied clients are less prone to visit health care facilities so that it contributes to maternal mortality [22].

Satisfaction with maternal care service is important both to the mother, infant and for the family which is evidenced by different studies showing mother's positive perception of birth experience has been linked to positive feelings toward her infant and adaptation to the mothering role [23, 24].

There are evidence showings determinants of client satisfaction on labor and delivery service in developed nations. Whereas there is lack of pooled estimate in Ethiopia [23–26]. Therefore, this study aimed to summarize the evidence of determinants of client satisfaction on labor and delivery service in Ethiopia. The summarized evidence obtained from this study helps the concerned bodies to identify existing gaps and propose strategies to increase client satisfaction in Ethiopia.

# Materials and methods

## Study protocol

The protocol for this systematic review and meta-analysis has been registered in the International Prospective Register of systematic reviews (PROSPERO). The methodology of this systematic review and meta-analysis was once developed via following the Preferred Reporting Items for Systematic Reviews and Meta-Analyses (PRISMA) **See S1 File**.

**Eligibility criteria.** The study participants have been postnatal mothers/women in Ethiopia. The investigators encompass observational studies that have been conducted at a facility/community setting in different parts of Ethiopia on the client/women/mother satisfaction on labor and delivery service. Besides, studies that were published after 2010 and were available till Dec 2019 and written in English have been eligible for this systematic review and meta-analysis.

Articles with irretrievable full texts, records with unrelated outcome measures and articles with missing or insufficient outcomes had been excluded. Reviews, commentaries, editorial, case series/reports, and patient stories had been excluded from the systematic review.

**Study setting.** Institutional /community-based studies considering satisfaction.

## Inclusion criteria

**Study design and period.** Observational studies reporting client/women/mother satisfaction on Labor and delivery service in Ethiopia published after 2010 and before Dec 2019 were considered. Only English language full-text reports were included.

## Sources of studies and searching strategies

The authors performed systematic literature searches from the authentic major electronic databases such as MEDLINE, PubMed, EMBASE, Emcare, CINAHL (EBSCOhost), Web of Science, Scopus, Poplin and different literature sources including Google Scholar. Also, the hand (manual) search of various repositories was performed to retrieve unpublished studies and gray literature. We used MeSH terms, key terms and search strings by extracting from the review questions for all the searches.

Advanced search strategies were applied in major databases to retrieve relevant findings closely related to client/women/mother satisfaction on labor and delivery service in Ethiopia. The search was conducted with the aid of carefully selected keywords and indexing terms.

The search strategy included "Factors" OR "Associate" OR "related factors "OR "determinants" OR "Predictors" AND "client satisfaction "OR "women satisfaction" OR "mother satisfaction" "AND "labor "AND " delivery" AND "Ethiopia". Both authors constructed the search

strings (RF and MB). The overall search result was compiled using EndNote X9 citation manager software **S2 File**.

## Selection of studies

All search results were exported to the EndNote X9 citation manager and duplicated studies were removed. Later, articles were screened through a careful reading of the title and abstract. The two authors screened and evaluated the studies independently.

The titles and abstracts of studies that mentioned the outcomes of the review were considered for further evaluation to be included in the systematic review and meta-analysis. Then the full-texts of the retained studies were further evaluated based on the quality of their objective, methods, participants/population, and key findings.

The authors (RF, KE, and MB) independently evaluated the quality of the studies included against the Joanna Briggs Institute (JBI) critical appraisal tool for cross-sectional studies checklists. In case of disagreement between the quality assessment results of the two authors, the differences were resolved by consensus for inclusion. The overall study selection process is presented using the PRISMA statement flow diagram "**Fig 1**".

## Data extraction and recording

Findings from the selected studies were extracted and stored using data extraction template prepared on Microsoft Word and then to Excel (2016), followed by extraction of important data based on study characteristics (Region, first author, year of publication, study design, and outcome of interest) by the two authors independently. For the prevalence of client satisfaction, we used unconverted proportional data to calculate the proportion/prevalence in percentage using OpenMeta software. Most of the determinants of the client satisfaction summary measures were done by the pooled odds ratio using Comprehensive meta-analysis (CMA) software.

## Risk of bias assessment

The methodological reputability and quality of the findings of the included studies were critically evaluated using the quality assessment tool for observational studies cross-sectional developed by the Joanna Briggs Institute (JBI). To ensure quality, the investigators searched for studies using a comprehensive strategy (electronic databases, and manual search) which included published and/or unpublished studies.

To minimize bias, the two authors independently screened the studies using clear objective eligibility criteria. Publication bias-was explored using visual inspection of the funnel plot. Additionally, Egger's regression test was carried out to check the statistical symmetry of the funnel plot.

## Critical appraisal of studies

The methodological reputability and quality of the findings of the included studies were critically evaluated using the quality assessment tool for observational studies developed by the Joanna Briggs Institute (JBI). The two authors (RF and KE) independently evaluated the quality of the studies. The mean score of the two authors was taken for a final decision. The differences in the inclusion of the studies were resolved by consensus. The cross-sectional observational studies checklist was graded out of 9 points. The included studies were evaluated against each indicator of the tool and categorized as high-, moderate-, and low quality. A high-quality score above 80%, moderate-quality between 60%-80%, and low-quality below 60%.

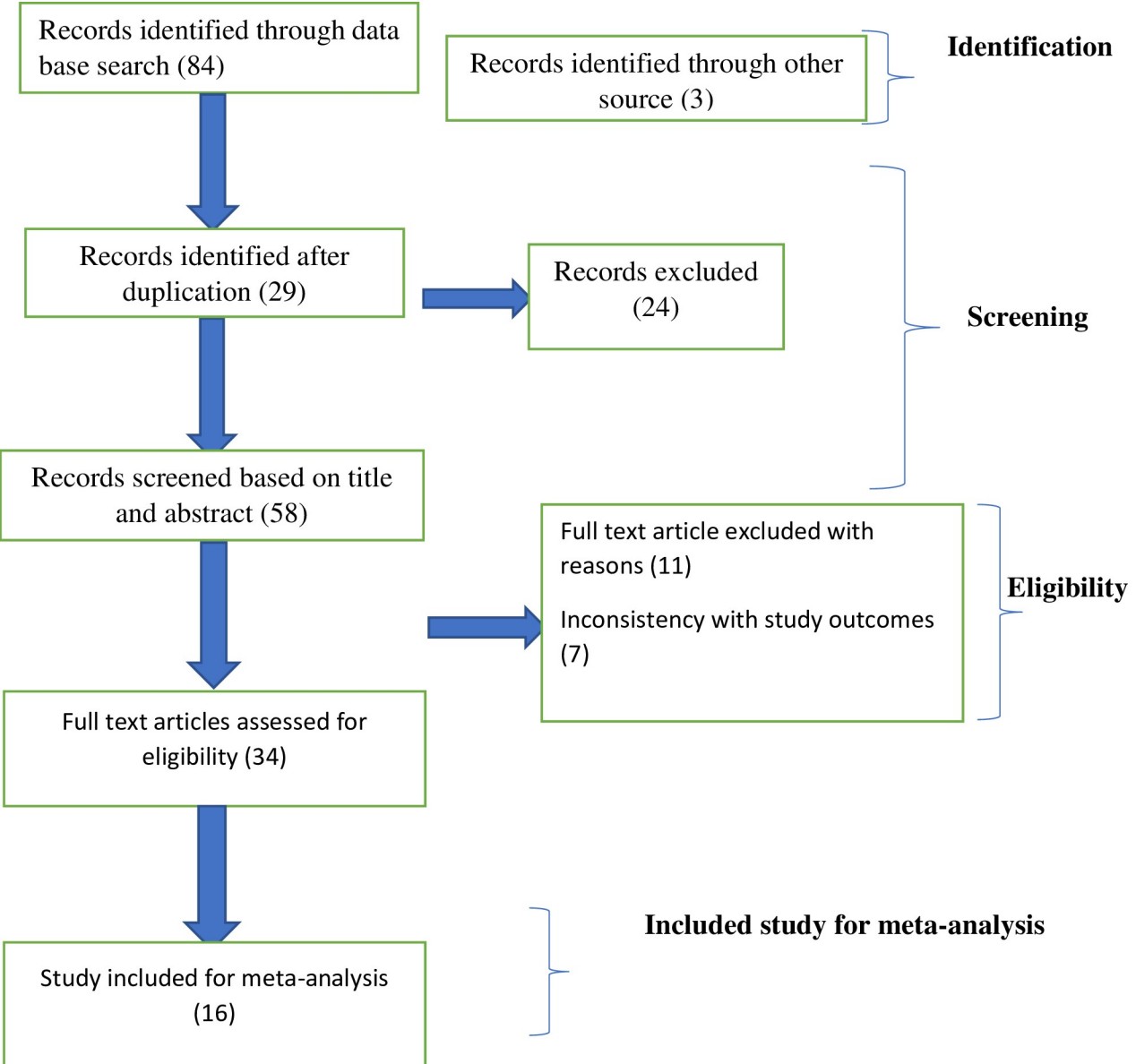

**Fig 1. Description of a schematic presentation of the PRISMA flow diagram to select and include studies, 2020.**

Studies with a score greater than or equal to 60% were included. This critical appraisal was conducted to assess the internal validity (systematic error) and external validity (generalizability) of studies and to reduce the risk of biases.

## Strategy for data synthesis

The findings of the included studies were first presented using a narrative synthesis. Study descriptions and summary (author-year, region, aim, design and population, sample size, and key findings) were compiled using Microsoft word. The raw numerical data prevalence (n) and total sample size (N)) from each study were extracted and recorded on a Microsoft word then exported to an Excel Spreadsheet. The authors conducted data synthesis and statistical analysis.

Meta-analysis was conducted using OpenMeta and CMA version 2 software to compute the pooled prevalence and determinates with client satisfaction on labor and delivery service.

A meta-analysis of observational studies was conducted, based on recommendations made by Higgins et al. (An $I^2$ of 75/100%, suggesting considerable heterogeneity). In the meantime, heterogeneity between the included studies was examined using the $I^2$ statistic.

Therefore, the presence of heterogeneity between studies was assumed if the $I^2$ statistic greater than seventy-five percent. A random-effects model was used to determine the pooled prevalence and determinates with client satisfaction on labor and delivery service in Ethiopia.

## Results

### Review studies

over-all of 87 articles have been identified through the main electronic databases and additional applicable sources search from January 1/2020 to Febraruary1/2/2020. From all identified studies, 29 articles were removed due to duplication 58 studies have been reserved for further screening. Of these, 24 have been excluded after being screened according to titles and abstracts. Of the 34 remaining articles, 18 studies had been excluded due to inconsistency with the inclusion criteria set for the review. Finally, 16 studies that fulfilled the eligibility criteria were included for the systematic review and meta-analysis. General characteristics and descriptions of the studies chosen for the meta-analysis have been outlined in "**Table 1**".

### Prevalence of client satisfaction on labor and delivery service

The pooled approximation of the magnitude of client satisfaction in Ethiopia was 73.5% (95% CI (64.9%, 82.1%) "**Fig 2**".

### Determinants of client satisfaction on labor and delivery service

The results of this review have shown determinants significantly associated with client satisfaction on labour and delivery service in Ethiopia, disrespectful maternal care (OR, 0.40;95% CI:0.19, 0.83, P = 0.01) Promptness of care (OR = 0.25; 95% CI: (0.18, 0.34) P = 0.0001), service charge (OR = 0.70; 95% CI: (0.57, 0.86) P < 0.0007), lack of privacy during examination (OR = 0.25; 95% CI: (0.10, 0.64) P < 0.004), not any plan to delivered at health facility (OR, 0.49; 95% CI: 0.37, 0.66, P < 0.00001) and no ANC follow-up (OR = 0.39; 95% CI: (0.24, 0.63) P < 0.0001) were less likely associated with client satisfaction on labor and delivery service in Ethiopia.

**Respectful maternal care.** The findings of the review indicated a significant association between respectful maternal care and client satisfaction. Disrespectful maternal care were 0.40 times less likely satisfied compared to respectful maternal care (OR = 0.40;95% CI: (0.19, 0.83) P = 0.01). Heterogeneity test indicated I2 = 75%, "**Fig 3**".

**Promptness of care.** The findings of the review indicated a significant association between delay to receive care and client satisfaction. Clients who wait more than one hour to receive care were 0.25 times less likely satisfied compared to clients who received care with less than an hour (OR = 0.25;95% CI: (0.18, 0.34) P = 0.0001). Heterogeneity test indicated I2 = 0%, "**Fig 4**".

**Service fee.** This review demonstrated that there was significant association between service charge and client satisfaction in the random model (OR = 0.70; 95% CI: (0.57, 0.86) P = 0.0007). Clients who were charged for the service were 0.70 times less likely satisfied as compared to clients who had received care free of charge "**Fig 5**".

**Table 1. Description of included studied for systematic review and meta-analysis 2020.**

| S. No | Authors, Year | Study setting | Study design | Data collection methods | Sample size | Prevalence % | Region | Out come | Specific factors |
|---|---|---|---|---|---|---|---|---|---|
| 1 | Blen Assefa, 2017 | Institutional | cross-section | Structured questioners | 461 | 82 | Addis Ababa administration | Maternal satisfaction on delivery service | Respectfulness of the staff, Mothers who received cognitive care, sex of the health workers who attended their deliveries, Mothers who felt like their babies received adequate care |
| 2 | Mesafint E, Worku A et al., 2014 | Institutional | cross-section | Structured questioners | 594 | 74.9 | Amhara region | Women satisfaction on child birth service | Age of women, antenatal care follow-up and the number of deliveries |
| 3 | Azmeraw T, Desalegn T et al., 2011 | Institutional | cross-section | Structured questioners | 417 | 61.9 | Amhara region | Women satisfaction on delivery service | Status of the pregnancy, immediate maternal condition after delivery, waiting time to see the health worker, availability of waiting area, care providers' measure taken to assure privacy during examinations, and amount of cost paid for service. |
| 4 | Kurabachew B, Mekonnen A, et al., 2015 | Community | cross-section | Structured questioners | 398 | 81.7 | Amhara region | Maternal satisfaction on delivery service | Having plan to deliver at health institution, waiting time. |
| 5 | Kiros T, Tsegaye AT et al., 2019 | Institutional | cross-section | Structured questioners | 593 | 31.3 | Amhara region | Client satisfaction on labor and delivery service | Travel time, mode of delivery, payment free delivery service |
| 6 | Gizew&Dessie Asres 2018 | Institutional | cross-section | Structured questioners | 420 | 88 | Amhara region | Maternal satisfaction on delivery service | level of education, access to ambulance service, welcoming hospital environment, proper pain management and listening to their questions. |
| 7 | Agegnehu B, Kedir R et al., 2019 | Institutional | Cross-sectional | Structured questioners | 398 | 84.7 | Harar | Women satisfaction on delivery service | a minimal waiting time to be seen by the healthcare provider, ample availability of emergency drugs within the hospital, not having antenatal care follow-up, having a previous experience of home delivery, planning to deliver in the hospital, and experiencing a short hospital stay after delivery |
| 8 | Roza A, Mesfin T, 2014 | Institutional | Cross-sectional | Structured questioners | 398 | 80.7 | Oromia region | Maternal satisfaction on delivery service | Age and educational level of the mother |
| 9 | AlemayehuG, Bosena T, 2018 | Institutional | Cross-sectional | Structured questioners | 366 | 78.7 | Oromia region | Maternal satisfaction on delivery service | planned delivery, free delivery service, perceived cleanness of toilets, and perceived presence of privacy and empathetic interactions of staffs |
| 10 | Biniyam H, Negalign B et al., 2017 | Institutional | Cross-sectional | Structured questioners | 391 | 65.2 | Oromia region | Maternal satisfaction on delivery service | ANC attendance, utilization of maternity waiting home (MWH) service, planned status of the pregnancy, distance and cleanliness of the toilet during delivery service |
| 11 | Aman U & Girum S, 2019 | Institutional | Cross-sectional | Structured questioners | 477 | 74.6 | Oromia region | Maternal satisfaction on delivery service | Educational status, economic status, privacy of mothers, mode of delivery, ANC attendance |

*(Continued)*

**Table 1.** (Continued)

| S. No | Authors, Year | Study setting | Study design | Data collection methods | Sample size | Prevalence % | Region | Out come | Specific factors |
|---|---|---|---|---|---|---|---|---|---|
| 12 | Teklemariam E, Wasihun A et al., 2019 | Institutional | Cross-sectional | Structured questioners | 280 | 30.4 | South region | Maternal satisfaction on delivery service | Antenatal care visit, health center delivery and less prolonged labor |
| 13 | MarishetA, ZemenuY et al., 2018 | Institutional | Cross-sectional | Structured questioners | 398 | 87.7 | South region | Maternal satisfaction on delivery service | being a student, instrumental delivery and waiting time to get obstetric care providers, getting immediate attention by providers |
| 14 | Abrham A, Amene A, et al., 2018 | Institutional | Cross-sectional | Structured questioners | 736 | 95 | South region | Maternal satisfaction on delivery service | Residence, unwanted pregenacy, overall cleaness, ANC attendance |
| 15 | Rahel T, Amare W et al., 2016 | Institutional | Cross-sectional | Structured questioners | 430 | 79.1 | South region | Maternal satisfaction on delivery service | The presence of support persons during child birth, women who delivered with caesarean section |
| 16 | Taklu M, Hinsermu B et al., 2017 | Institutional | Cross-sectional | Structured questioners | 413 | 79.7 | Tigray region | Client satisfaction on labor and delivery service | Respondents live in rural area, stayed<4 days, admitted for the first time, service charge |

**Privacy.**   Privacy maintained during the examination was significantly associated with client satisfaction, the odds of client satisfaction were high among those who maintained privacy as compared to others (OR = 0.25; 95% CI: (0.10, 0.64) P = 0.004). Clients who lack privacy were 0.25 times less likely satisfied as compared to those who maintained privacy.

Considerable heterogeneity was found too high ($I^2$ = 92%), hence the random effect model was assumed in the analysis. Sensitivity was done of analysis but did not bring significant change in the overall summary results of OR "**Fig 6**".

**ANC follow-up.**   The findings of the review indicated a significant association between ANC follow-up and client satisfaction. Clients with no ANC follow-up were 0.39 times less likely satisfied compared to ANC follow-up (OR = 0.39;95% CI: (0.24, 0.63) P = 0.0001). Heterogeneity test indicated I2 = o%, "**Fig 7**".

**Plan to deliver at the facility.**   The results of the review presented there was statistically significant association between clients who had the plan to deliver at the health facility and client satisfaction (OR = 0.49; 95% CI: (0.37, 66) P<0.00001). Clients who had no plan to deliver at the health facilities were 0.49 times less likely satisfied than their counterparts "**Fig 8**".

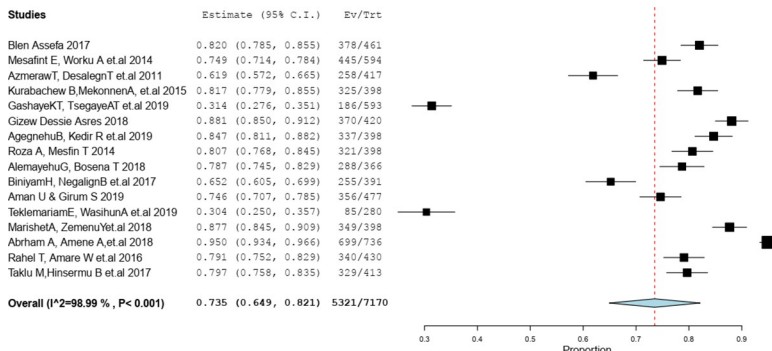

**Fig 2. Pooled prevalence of client satisfaction on labor and delivery service in Ethiopia, 2020.**

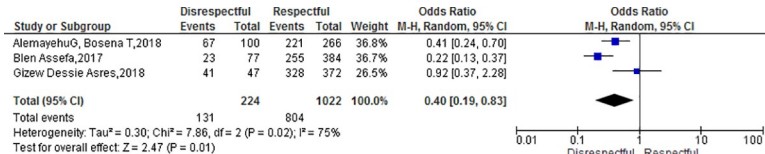

**Fig 3. Association between respectful maternal care with client satisfaction in Ethiopia, 2020.**

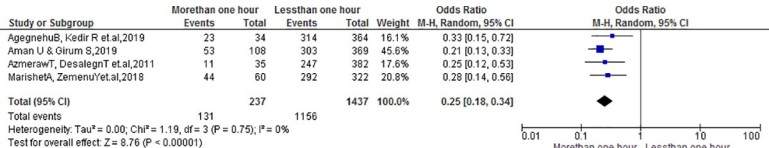

**Fig 4. Association between delay to receive care with client satisfaction in Ethiopia, 2020.**

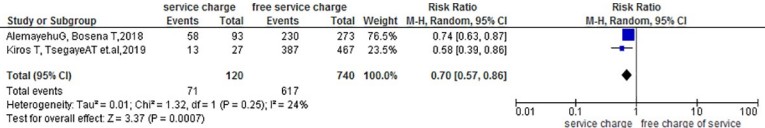

**Fig 5. Association between service charge with client satisfaction in Ethiopia, 2020.**

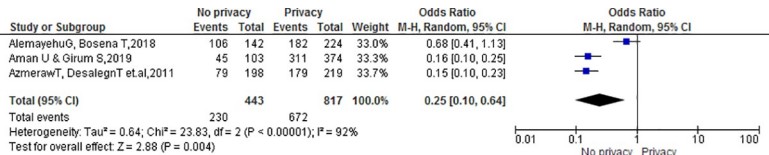

**Fig 6. Association between privacy with client satisfaction in Ethiopia, 2020.**

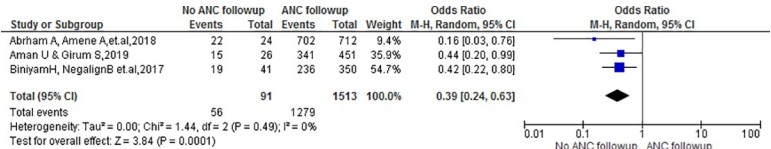

**Fig 7. Association between ANC follow-up with client satisfaction in Ethiopia, 2020.**

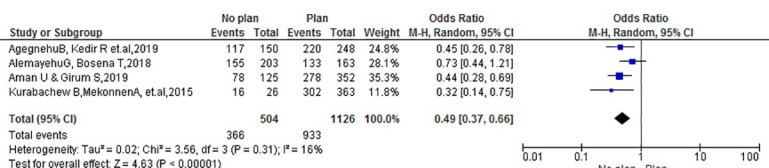

**Fig 8. Association between plan to deliver at a health facility with client satisfaction in Ethiopia, 2020.**

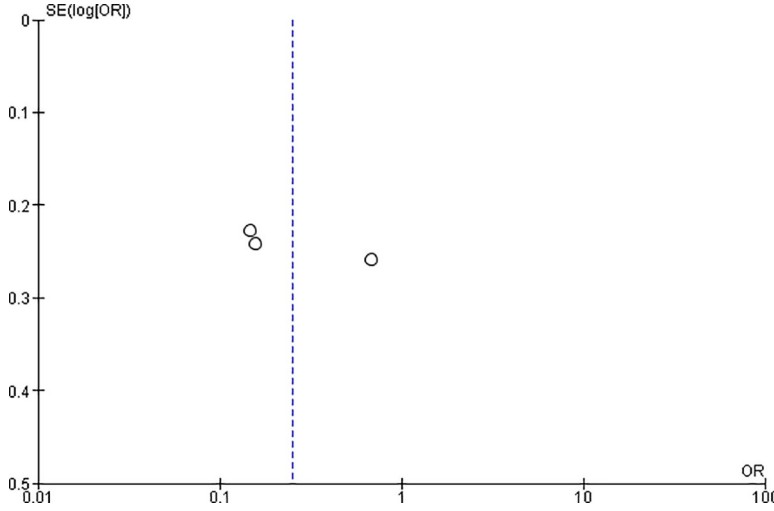

**Fig 9. Publication bias on the privacy of the clients during an examination.**

## Publication bias

To check publication bias among the included studies for the meta-analysis, funnel plot and Egger's test were carried out. "**Figs 9 and 10**".

## Discussion

Evaluating client satisfaction remains one of the key performance indicators to control the quality of services within a facility which provides maternal care. This comprehensive study provides potted information on determinants that affect client satisfaction on labor and delivery service in Ethiopia. This review incorporated studies with methodological quality and all of the studies were descriptive cross-sectional.

This review revealed that the pooled prevalence of client satisfaction on labor and delivery service in Ethiopia was 73.5%. The pooled odds ratio showed a negative association between client satisfaction on labour and delivery service with Promptness of care. Free service charge,

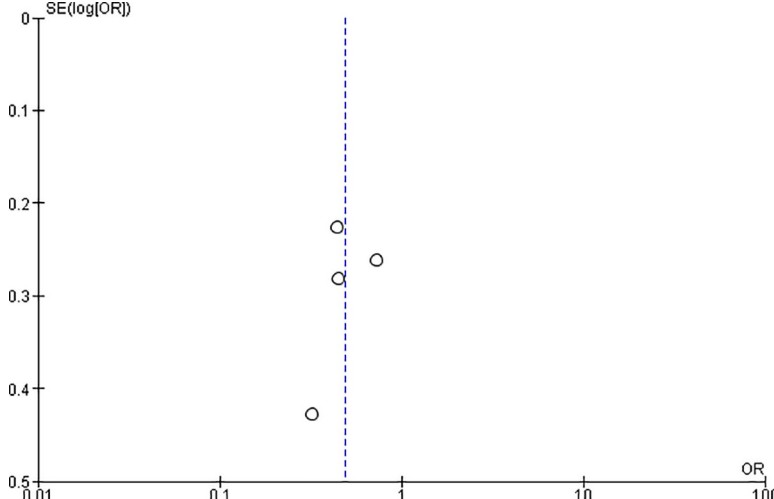

**Fig 10. Publication bias on a plan for delivery at a health facility.**

Privacy during examination, Respectful maternal care, Plan to delivered at health facility and ANC follow-up were factors positively associated with client satisfaction on labor and delivery service in Ethiopia.

Evaluating client satisfaction remains one of the key performance indicators to control the quality of services within a facility which provides maternal care. This comprehensive study provides potted information on determinants that affect client satisfaction on labor and delivery service in Ethiopia. This review incorporated studies with methodological quality and all of the studies were descriptive cross-sectional.

This systematic and meta-analysis revealed that the overall prevalence of client satisfaction on labor and delivery service during the period studied in Ethiopia was 73.5% (95 CI: 64.9 82.1). This result is highest than studies in Eritrea 20.8% [27] and Kenya 54.5% [19]. The possible reason for this might be due to policy difference that all maternal service was free of charge, competency of providers and the expansion of training corresponding EMONC, companionate and respectful care for MCH providers. However, client satisfaction in our meta-analysis was lower than the study done in Nepal 86% [28] and Ghana 80% [29]. The difference might be due to poor job satisfaction of providers, due to an inadequate number of providers.

Our analysis revealed that the magnitude of client satisfaction across the region shows few discrepancies in Oromia 74.8%, Southern Nations Nationalities and Peoples Region (SNNPR) 73.1% and in Amhara 67.6%. The possible reason for this discrepancy due to the difference in the standard of care, the difference in exercising compassionate and respectful maternal care and might be due to the presence of experienced providers. In this review and meta-analysis, we found several determinants that had a significant association with client satisfaction on labor and delivery service in Ethiopia. The current review indicated that National policy, structural and process play a significant role in client satisfaction on labor and delivery service. This meta-analysis shows that client satisfaction on labor and delivery service was positively associated with, respectfulness of providers, promptness of care, privacy, plan to deliver at the facility, free service charge, and ANC follow-up.

This review revealed that the client who received respectful maternal care by providers was positively associated with client satisfaction on labor and delivery [30–32]. A similar result was seen from another primary study in Zambia, Malawi, Vietnam, Oman, and southeast Nigeria and systematic review in developing countries, where client satisfaction on labor and delivery were positively associated with respectfulness of providers [23, 33–38]. This is due to, clients treated based on the principle of women-centered approaches, by curiosity, empathy so that it can allow exercising their right and to receive a high level of quality of services. In contrast a study in Pakistan, Turkey and Ghana it was revealed that the absence of a respectful approach by providers increased the odds of dissatisfaction of clients on labor and delivery service [39–41].

In this review privacy during examination increased client satisfaction [32, 42, 43]. Similarly, systematic review in developing country reported that privacy during examination increases client satisfaction on labor and delivery service [37]. The possible reason for this might be due to norm of the people and keeping privacy avoids shame and it builds confidence. A study in Eritrea, Kenya, Nepal, Bangladesh, and India revealed a lack of privacy increased the odds of dissatisfaction of clients on labor and delivery service [19, 27, 28, 44, 45].

Besides, free service charge was among the determinants contributing to client satisfaction on labor and delivery service in Ethiopia [32, 46]. The finding of this review is consistent with studies conducted in Kenya, Nepal, and Gambia [19, 28, 47]. This might be due to poor socio-economic status of clients. This facilitates room to use the money for other needs that may arise during their pregnancy-child birth process. In contrast a study in India revealed that service charge for labor and delivery service was positively associated with client satisfaction [45].

The review revealed that promptness of care was associated with client satisfaction [42, 43, 46, 48, 49]. This was in line with a study in Oman, Nigeria, Bangladesh and India and systematic review in developing countries [23, 35, 37, 50].

In this review ANC follow-up was associated with client satisfaction [43, 51–54]. The possible reason might be that the clients will have a repeated exposure to the environment which may indicate provision of good quality of care resulting in satisfaction.

The plan to delivered at the health facility was positively associated with client satisfaction [32, 43, 46, 55, 56]. The possible reason for this might be based on free transportation/ambulance access, compassionate and respectful care received during the antepartum period and pre-planned scheduled for every expected outcome.

## Conclusions

This review revealed that client satisfaction on labor and delivery service in Ethiopia was 73.5%. Besides poor care of providers on the antepartum, intrapartum and lack compassionate and respectful care affects client satisfaction on labor and delivery service in Ethiopia. To enhance client satisfaction on labour and delivery service integrated effort is a need within, minister of health and with respective stakeholders by designing continuous monitoring and evaluation system of the health care system and arranging training for providers on compassionate and respectful care. Promptness of care, cost of the service, respectfulness of the providers, privacy, plan to deliver at the health facility and ANC follow-up were factors associated with client satisfaction on labor and delivery service. This meta-analysis has certain limitation, all of the studies included were cross-sectional so that it restricts us to do cause-effect analysis. The included studies have difference in the quality of study which is observed by heterogeneity.

## Supporting information

**S1 File. PRISMA 2009 checklist.**
(DOC)

**S2 File.**
(DOCX)

**S3 File.**
(XLSX)

## Acknowledgments

We would like to thank the College of medicine and health science, department of midwifery for non-financial support.

## Author Contributions

**Conceptualization:** Rekiku Fikre, Muntasha Berhanu.

**Formal analysis:** Rekiku Fikre, Kidist Eshetu, Akalewold Alemayehu.

**Investigation:** Muntasha Berhanu.

**Methodology:** Rekiku Fikre, Kidist Eshetu.

**Resources:** Kidist Eshetu, Muntasha Berhanu.

**Software:** Rekiku Fikre.

**Writing – original draft:** Rekiku Fikre, Kidist Eshetu, Muntasha Berhanu.

**Writing – review & editing:** Rekiku Fikre, Muntasha Berhanu, Akalewold Alemayehu.

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
