## [Decision Letter · Decision Letter 0]

13 Oct 2020

PONE-D-20-11487

What determines client satisfaction on labor and delivery service in Ethiopia? systematic review and meta-analysis

PLOS ONE

Dear Dr. Abebe,

Thank you for submitting your manuscript to PLOS ONE. After careful consideration, we feel that it has merit but does not fully meet PLOS ONE’s publication criteria as it currently stands. Therefore, we invite you to submit a revised version of the manuscript that addresses the points raised during the review process.

We look forward to receiving your revised manuscript.

Kind regards,

Onikepe Oluwadamilola Owolabi

Academic Editor

PLOS ONE

Journal Requirements:

2. Please confirm that you have included all items recommended in the PRISMA checklist including details of reasons for study exclusions in the PRISMA flowchart and number of studies excluded for each reason, the rationale for the search start date, and the PROSPERO registration number.

<h3>** **</h3>

4. Thank you for stating the following financial disclosure: 'NO'

*Please include your amended statements within your cover letter; we will change the online submission form on your behalf.*

5. Thank you for stating the following in your Competing Interests section: 'NO'

a. Please complete your Competing Interests statement to state any Competing Interests. If you have no competing interests, please state "The authors have declared that no competing interests exist.", as detailed online in our guide for authors at http://journals.plos.org/plosone/s/submit-now

 b.This information should be included in your cover letter; we will change the online submission form on your behalf.

7. Please upload a copy of Figure 1, to which you refer in your text on page 7. If the figure is no longer to be included as part of the submission please remove all reference to it within the text.

8. Please include a copy of Table 1 which you refer to in your text on page 10.

9. Please include captions for your Supporting Information files at the end of your manuscript, and update any in-text citations to match accordingly. Please see our Supporting Information guidelines for more information: http://journals.plos.org/plosone/s/supporting-information

Reviewers' comments:

Reviewer's Responses to Questions

**Comments to the Author**

1. Is the manuscript technically sound, and do the data support the conclusions?

Reviewer #1: Partly

2. Has the statistical analysis been performed appropriately and rigorously? 

Reviewer #1: Yes

3. Have the authors made all data underlying the findings in their manuscript fully available?

Reviewer #1: Yes

4. Is the manuscript presented in an intelligible fashion and written in standard English?

Reviewer #1: No

5. Review Comments to the Author

Reviewer #1: Abstract – would like to see it formatted a bit more clearly, perhaps leaving at least one line between the sections so that it is easier to read:

• Background

• Methods

• Results

• Conclusion

Background – there are sentences that do not make sense, please see highlights in yellow in the manuscript

Methods – when were the databases searched?

Conclusion – is a repeat of the results, the conclusion should answer the question, “So what?”. You have obtained the results that you have obtained. So what should we know or do about this?

Please see atttached WORD file for more comments

6. PLOS authors have the option to publish the peer review history of their article (what does this mean?). If published, this will include your full peer review and any attached files.

Reviewer #1: No

---

## [Author Response · Author response to Decision Letter 0]

2 Dec 2020

We appreciate the editor and the reviewer for the time and effort dedicated to review and give feedback on the manuscript.

We have tried to incorporate all the suggestions given by the reviewers and we have made modification of the draft manuscript on track change. The point by point response to the reviewers’ concerns and editor comments are uploded in separate file.

---

## [Decision Letter · Decision Letter 1]

18 Feb 2021

PONE-D-20-11487R1

What Determines Client Satisfaction on Labor and Delivery service in Ethiopia? Systematic review and Meta-analysis

PLOS ONE

Dear Dr. Abebe,

Thank you for submitting your manuscript to PLOS ONE. After careful consideration, we feel that it has merit but does not fully meet PLOS ONE’s publication criteria as it currently stands. Therefore, we invite you to submit a revised version of the manuscript that addresses the points raised during the review process.

We look forward to receiving your revised manuscript.

Kind regards,

Onikepe Oluwadamilola Owolabi

Academic Editor

PLOS ONE

Reviewers' comments:

Reviewer's Responses to Questions

**Comments to the Author**

1. If the authors have adequately addressed your comments raised in a previous round of review and you feel that this manuscript is now acceptable for publication, you may indicate that here to bypass the “Comments to the Author” section, enter your conflict of interest statement in the “Confidential to Editor” section, and submit your "Accept" recommendation.

Reviewer #1: (No Response)

2. Is the manuscript technically sound, and do the data support the conclusions?

Reviewer #1: Yes

3. Has the statistical analysis been performed appropriately and rigorously? 

Reviewer #1: Yes

4. Have the authors made all data underlying the findings in their manuscript fully available?

Reviewer #1: Yes

5. Is the manuscript presented in an intelligible fashion and written in standard English?

Reviewer #1: No

6. Review Comments to the Author

Reviewer #1: I think this is an important manuscript which should be published, and I believe the authors have put a lot of work and effort into the document.

However there are grammatical and spelling mistakes, sometimes words - verbs - are missing in sentences.

I would It would be unfortunate if the manuscripot on this very important topic could not be published because of these reasons.

I would therefore urge the authors to go through the manuscript again and check it thoroughly for spelling and grammatical mistakes. Please state what acronymns mean the first time you use them.

Perhaps you can identify a native English speaker to help you?

7. PLOS authors have the option to publish the peer review history of their article (what does this mean?). If published, this will include your full peer review and any attached files.

Reviewer #1: **Yes: **Owolabi Bjälkander

---

## [Author Response · Author response to Decision Letter 1]

6 Mar 2021

Manuscript Number: PONE-D-20-11487R1

Response to the Academic editors and reviewers

Respected 

Onikepe Oluwa Damilola Owolabi

We would like to thank for the opportunity we get again to submit the revised version of the manuscript entitled “What Determines Client Satisfaction on Labor and Delivery Service in Ethiopia: Systematic Review and Meta-Analysis” to publish on Plose one. We appreciate you and the reviewer for the time and effort dedicated to review and give feedback on the manuscript. We critically observed the comments by the academic editors and reviewers, and we observed that the majority of the comments and concerns were addressed on the previous revisions and now we gave much emphasis on grammar and spelling errors which was raised by the reviews. So that we have tried to incorporate all the suggestions given by the academic editors and reviewers and we have made revision of the manuscript on track change. The point-by-point response to the reviewers’ concerns and comments are presented here below. 

Point by point response for the comments from the academic editors and reviewer on manuscript entitled “What Determines Client Satisfaction on Labor and Delivery Service in Ethiopia: Systematic Review and Meta-Analysis” 

1. Reviewer's Responses to Questions

Comments to the Author

If the authors have adequately addressed your comments raised in a previous round of review and you feel that this manuscript is now acceptable for publication, you may indicate that here to bypass the “Comments to the Author” section, enter your conflict-of-interest statement in the “Confidential to Editor” section, and submit your "Accept" recommendation. 

Reviewer #1: (No Response)

Authors response 

We appreciate the feedback from of our academic editors and reviewers, and we incorporate all the points raised by our reviewers.

2. Reviewer's Responses to Questions

Comments to the Author

Is the manuscript technically sound, and do the data support the conclusions?

Reviewer #1: Yes

Authors response 

We saw the response and thankfull

3. Reviewer's Responses to Questions

Comments to the Author

Has the statistical analysis been performed appropriately and rigorously?

 Reviewer #1: Yes

Authors response 

We saw the response and thankfull

4. Reviewer's Responses to Questions

Comments to the Author

Have the authors made all data underlying the findings in their manuscript fully available?

Reviewer #1: Yes

Authors response 

We saw the response and thankfull



5. Reviewer's Responses to Questions

Comments to the Author

Is the manuscript presented in an intelligible fashion and written in standard English?

Reviewer #1: No

Authors response to the academic editors and reviewer

We accepted the comment and inccorporated all the possible corections.

Review Comments to the Authors

Reviewer #1: I think this is an important manuscript which should be published, and I believe the authors have put a lot of work and effort into the document. However there are grammatical and spelling mistakes, sometimes words - verbs - are missing in sentences.

I would It would be unfortunate if the manuscripot on this very important topic could not be published because of these reasons.I would therefore urge the authors to go through the manuscript again and check it thoroughly for spelling and grammatical mistakes. Please state what acronymns mean the first time you use them.Perhaps you can identify a native English speaker to help you?

Authors response 

We postively accepted the comment and invited two nativespeakers for gramer and editorial task and inccorporated all the possible corections.

---

## [Editor Report · Decision Letter 2]

30 Mar 2021

What Determines Client Satisfaction on Labor and Delivery service in Ethiopia? Systematic review and Meta-analysis

PONE-D-20-11487R2

Dear Dr. Abebe,

We’re pleased to inform you that your manuscript has been judged scientifically suitable for publication and will be formally accepted for publication once it meets all outstanding technical requirements.

Kind regards,

Onikepe Oluwadamilola Owolabi

Academic Editor

PLOS ONE
---

## [Editor Report · Acceptance letter]

5 Apr 2021

PONE-D-20-11487R2 

What Determines Client Satisfaction on Labor and Delivery Service in Ethiopia? Systematic review and Meta-analysis 

Dear Dr. Fikre:

I'm pleased to inform you that your manuscript has been deemed suitable for publication in PLOS ONE. Congratulations! Your manuscript is now with our production department. 

Kind regards, 

on behalf of

Dr. Onikepe Oluwadamilola Owolabi 

Academic Editor

PLOS ONE